# Thyroid Hormone Receptor Isoforms Alpha and Beta Play Convergent Roles in Muscle Physiology and Metabolic Regulation

**DOI:** 10.3390/metabo12050405

**Published:** 2022-04-29

**Authors:** Annarita Nappi, Melania Murolo, Annunziata Gaetana Cicatiello, Serena Sagliocchi, Emery Di Cicco, Maddalena Raia, Mariano Stornaiuolo, Monica Dentice, Caterina Miro

**Affiliations:** 1Department of Clinical Medicine and Surgery, University of Naples “Federico II”, 80131 Naples, Italy; annarita.nappi@unina.it (A.N.); mel.murolo@gmail.com (M.M.); nunziacicatiello3011@gmail.com (A.G.C.); ssagliocchi@gmail.com (S.S.); emery2304@gmail.com (E.D.C.); monica.dentice@unina.it (M.D.); 2CEINGE—Biotecnologie Avanzate S.c.a.r.l., 80131 Naples, Italy; raia@ceinge.unina.it; 3Department of Pharmacy, University of Naples “Federico II”, 80149 Naples, Italy; mariano.stornaiuolo@unina.it

**Keywords:** thyroid hormones (THs), thyroid hormone receptors (TRs), skeletal muscle, muscle metabolism

## Abstract

Skeletal muscle is a key energy-regulating organ, skilled in rapidly boosting the rate of energy production and substrate consumption following increased workload demand. The alteration of skeletal muscle metabolism is directly associated with numerous pathologies and disorders. Thyroid hormones (THs) and their receptors (TRs, namely, TRα and TRβ) exert pleiotropic functions in almost all cells and tissues. Skeletal muscle is a major THs-target tissue and alterations of THs levels have multiple influences on the latter. However, the biological role of THs and TRs in orchestrating metabolic pathways in skeletal muscle has only recently started to be addressed. The purpose of this paper is to investigate the muscle metabolic response to TRs abrogation, by using two different mouse models of global TRα- and TRβKO. In line with the clinical features of resistance to THs syndromes in humans, characterized by *THRs* gene mutations, both animal models of TRs deficiency exhibit developmental delay and mitochondrial dysfunctions. Moreover, using transcriptomic and metabolomic approaches, we found that the TRs–THs complex regulates the Fatty Acids (FAs)-binding protein GOT2, affecting FAs oxidation and transport in skeletal muscle. In conclusion, these results underline a new metabolic role of THs in governing muscle lipids distribution and metabolism.

## 1. Introduction

Thyroid hormones (THs), L-Thyroxine (T4, or 3,3′,5,5″-tetraiodo-L-thyronine) and the physiologically active form Triiodothyronine (T3, or 3,3′,5-triiodo-L-thyronine), are crucial determinants of development, tissue differentiation, and energy homeostasis maintenance [1,2]. THs act in various tissues by regulating the transcription of specific target genes [3,4,5] through genomic and non-genomic action. The ultimate effects of THs mostly depend on tissue T3 bioavailability and the presence of key regulators of TH signaling including: (1) the plasma membrane transporters, among which MCT8 and MCT10 have been widely studied; (2) the differential expression of THs Receptor isoforms (TRs) (TRα and TRβ, encoded by *THRα* and *THRβ* genes, respectively); and (3) the activity of three specific seleno-enzymes able to modify the intracellular THs signaling, namely, the type 1, type 2 and type 3 deiodinases (D1, D2 and D3) [6,7]. In particular, the systemic concentrations of THs are quite stable and controlled by the Hypothalamic–Pituitary–Thyroid (HPT) axis, but THs levels in the plasma do not faithfully reflect their availability in cells, where TH signaling is modified via the concerted action of D1, D2 and D3. Indeed, the deiodinases regulate the peripheral metabolism of THs, by catalyzing the removal of iodine moieties at different sites of the phenolic (5′ position) or tyrosyl (5 position) ring of THs, respectively catalyzing their activation (mediated by D1 and D2) and inactivation (mediated by D3) [6,8]. Although in the last decade, a large number of non-genomic actions of THs have been described [9,10,11,12,13], the most characterized effects of THs refer to their genomic action, which consists in modulating gene transcription through the binding to nuclear TRs [14]. Upon ligand binding, the TRs act as a homo- or heterodimer in association with Retinoic Acid Receptors (RXRs) and bind to specific DNA regions located in the regulatory regions of target genes, called Thyroid Response Elements (TREs), recruiting chromatin remodeling complexes, which modify histones leading to dynamic interchange between the open (transcriptionally active) and closed (transcriptionally repressed or silenced) chromatin state [15,16]. The expression of the different TRs isoforms, TRα1, TRα2 and TRβ1, varies in different tissues and cell types, based on specific temporal and spatial patterns during development [17]. While the TRα isoforms are expressed predominantly in brain, heart, skeletal muscle and adipose tissues, the TRβ isoform is expressed in liver, heart and pituitary, and mediates the regulation of cholesterol metabolism, as well as the feedback regulation of Thyrotropin-Stimulating Hormone (TSH) in the pituitary [18]. Skeletal muscle is an important THs-target tissue and an example of the contextual expression of both TRs isoforms, with the TRα acting as a primary receptor, involved in the regulation of proliferation and differentiation of myoblasts [19]. 

In the past decades, different studies have revealed that intracellular modulation of THs action, mediated by the deiodinases D2 and D3 and the TRs, plays a crucial role in the regulation of the myogenic progression, regeneration, and metabolism of skeletal muscle. THs stimulate the expression of Myosin Heavy Chain (MHC) characteristic of fast-twitch fibers and increase mitochondrial biogenesis and the relaxation–contraction rate. Crucial for the progression of muscle progenitor cell differentiation is the intracellular THs concentration, which is finely regulated by the combined activity of D2 and D3 [7,20]. In particular, the intracellular THs concentration should be maintained at a low level at the beginning of the myogenic process [21,22]. Indeed, D3 is highly expressed in activated and proliferating Satellite Cells (SCs), and it is downregulated during the differentiation process [22], when instead D2 is upregulated, leading to an increase in intracellular THs concentration that drives the terminal differentiation of myocytes into myotubes/myofibers. 

Animal models with an alteration in THs signaling showed different and specific skeletal muscle phenotypes. In detail, changes in contractile and metabolic proprieties of the muscle fibers have been described in hyper- and hypothyroidism conditions. Moreover, most patients with hypothyroidism have been found myopathic changes, including muscle weakness and pseudohypertrophy, myasthenic syndrome and rhabdomyolysis, such as the onset of muscle weakness and atrophy in hyperthyroid patients [23]. On the other hand, different degrees of muscle weakness and atrophy are also well known in hyperthyroid patients [24].

Mouse models of global TRα- and TRβ- knock-out (KO) have been generated and widely described [25,26]. In detail, TRαKO mice (due to the disruption of both TRα1 and TRα2 isoforms) display progressive hypothyroidism, with clear growth retardation together with multiple disorders, such as a lower body temperature, a strongly delayed maturation of bone and intestine, thus leading to death shortly after the weaning period [25,27]. This phenotype is in agreement with the phenotype observed in patients with mutations in the *THRα* gene, which are characterized by highly variable phenotypic features of hypothyroidism with skeletal dysplasia, poor growth, neurodevelopmental retardation, low metabolic rate and constipation [28,29,30].

Strikingly, the TRβKO mice overproduce THs and TSH, in agreement with the concept that the TRβ receptors are the most potent homeostatic regulators of the production of TSH [31]. Moreover, the inactivation of the *THRβ* gene in mice results in impairment of the auditory function, but no alteration in development, metabolism or neurological functions [26,31]. The mice model lacking both TRs isoforms (TRα/β KO) exhibits a dramatic change in phenotype, including poor female fertility, hyperactivity of the pituitary–thyroid axis, and retardation of growth and bone development [32]. 

Although it has been clearly established that THs and their receptors are essential for embryonic and post-natal development, their precise functions in the physiology of each single tissue and skeletal muscle have been only partially elucidated. 

The aim of our work was to characterize the phenotype of skeletal muscles in TRα- and TRβKO mice, with a particular focus on the metabolic alterations caused by TRs Loss-Of-Function (LOF), to expand the knowledge regarding the long-recognized “thyroid hormone–metabolism connection” [33]. Indeed, a large body of literature demonstrates that the increase in THs production (hyperthyroidism) or exogenous administration of THs potently raises the metabolic rate, while THs deficiency induces a hypo-metabolic state accompanied by a reduction in metabolic rate and energy expenditure [33,34].

## 2. Results

### 2.1. Phenotypical Analysis of TRα- and TRβKO Muscles

To study the relevance of TRs in skeletal muscle physiology, we used *THRα*- and *THRβ*- LOF mouse models [27,31], here referred to as TRα- and TRβKO, respectively. Compared to control (CTR) mice, TRα- and TRβKO mice showed a significant reduction in body weight and skeletal muscle mass (Figure 1A–E); although, the proportion of visceral (Vi-WAT) and subcutaneous (Sub-WAT) white adipose tissues were unchanged between genotypes (Figure 1F,G). From a morphometric perspective, the mean Cross-Sectional Area (CSA) of the Tibialis Anterior (TA) muscles of TRβKO mice was similar in appearance to the CSA in CTR mice, while TRαKO TA muscles showed small-caliber skeletal muscle fibers compared to CTR (Figure 1H). 

These results are in agreement with the role of THs and their receptors in muscle development and with the growth retardation of mice and humans with *THRα* and *THRβ* mutations [32].

### 2.2. THRα and THRβ Deficiency Affects Mitochondrial Dynamics and Function

To characterize the effect of TRs alteration on mitochondrial content and energy metabolism, first, we performed succinate dehydrogenase (SDH) and nicotinamide adenine dinucleotide (NADH) staining of TA histological sections. In detail, the evaluation of the NADH and SDH staining was performed visually, by dividing the muscle fibers into three groups, named light, intermediate and dark, based on the intensity of their staining. Quantification of metachromatic stain intensities revealed an enhanced Complex I (measured by NADH, Figure 2A, bottom row) and Complex II (measured by SDH, Figure 2A, top row) activities in TRα- and TRβKO muscles compared to CTR, which reflect an alteration in mitochondrial bioenergetics and functionality. Next, we measured the expression levels of several mitochondrial markers. mRNA expression of different mitochondrial regulatory factors involved in fusion and fission processes (MFN1, MFN2, OPA1 and DRP1) was significantly reduced in TRα- and TRβKO mice compared to CTR, as well as the expression of PGC1-α, a marker of mitochondriogenesis, and UCP3, which were down-regulated in both TRα- and TRβKO mice (Figure 2B). 

Analysis of the mitochondrial DNA copy number (mtDNA-CN), assessed by quantitative Real-Time PCR, revealed that the mtDNA-CN in TRαKO muscles is unchanged compared to CTR, while in TRβKO muscles the mtDNA-CN is even higher than CTR (Figure 2C). Nevertheless, measurement of the mitochondrial membrane potential by Mito-Tracker staining on whole-muscle, demonstrated that both TRα- and TRβKO muscles have a lower mitochondrial function compared to CTR muscles (Figure 2D). Furthermore, analysis of mitochondrial production of superoxides, assessed by Mito-Sox staining on whole-muscle, revealed that TRα- and TRβKO muscles showed higher mitochondrial Reactive Oxygen Species (mt-ROS) production than CTR muscles (Figure 2E). Similar results were obtained in EDL-derived single muscle fibers isolated from CTR, TRα- and TRβKO mice (Figure 3A,B). Overall, these results demonstrate that TRs isoforms, TRα and TRβ, play a convergent role in the regulation of mitochondrial energy metabolism. Indeed, our results demonstrated that the THs–TRs complex regulates the mitochondrial function by affecting mitochondrial biogenesis and turnover, as well as the mitochondrial functionality and mt-ROS production, while the *THRα-* and *THRβ*-LOF conditions lead to mitochondrial dysfunction.

### 2.3. Loss of Both TRα and TRβ Isoforms Affects the Glutamine Metabolism in Skeletal Muscle

Besides the mitochondrial alterations observed in the muscles of TRα- and TRβKO mice, we also found a reduction in glutamine metabolism, which is consistent with our previous finding that both TRs isoforms contribute to mitochondrial glutamate aminotransferase (GPT2) expression, recently identified as novel TH-target gene in muscle cells and tissues [35]. To gain insight into the role of TRs in glutamine metabolism, we assessed the expression of several genes involved in glutamine metabolism regulation. Among these, Glutamate Pyruvate Transaminase isoforms (GPT and GPT2) mRNA expressions were significantly down-regulated in TRα- and TRβKO mice, as well as Glutamic Oxaloacetic Transaminase (GOT1 and GOT2), Glutamine Synthase (GS), Glutaminase (GLS), Glutamate Transaminase (GLUD) and glutamine transporters (SLC1A5, SLC6A14, SLC6A19, SLC7A5 and SLC7A8) (Figure 4). 

Consistent with our previous study [35], the above results suggest that the muscle-specific expression level of GPT2 and several glutamine metabolism-related genes are significantly reduced in both TRs-deficient mice models, confirming that THs modulate the glutamine metabolism and that GPT2 is essential for their pro-anabolic function.

### 2.4. Skeletal Muscle TRs Deficiency Impacts on Lipids Composition

Given the above-mentioned alterations in mitochondrial function observed in TRα- and TRβKO mice, we performed a global metabolomic analysis of TRα- and TRβKO GC muscles. Among the altered metabolites, lipids were the most significantly variated molecules in our analysis. Indeed, the Principal Component Analysis (PCA) showed that both TRα- and TRβ-deficient muscle metabolite compositions were similarly significantly altered compared to CTR. When we plotted the first three principal components of the PCA analysis of each tissue sample, which account for approximately 60% of the variability in the data, TRα-, TRβKO and CTR groups were completely segregated (Figure 5A,B). Thus, PCA analysis and dendrograms confirmed that there are significant global metabolic changes in muscle tissue, as a result of TRs deficiency (Figure 5A–D). We examined the changes in global lipidomic profiles in TRα- and TRβKO muscle compared to CTR and we identified 25 different lipidic metabolites (Figure 5C,D and Table 1). When ranked by FAs content, the statistical test of means (mixed one-way ANOVA, *p*-value < 0.05) performed between the TRα- and TRβKO groups compared to CTR, showed significant differences in seven compounds, namely, oleic acid (MUFA, 18:1), palmitoleic acid (MUFA, 16:1), palmitic acid (SFA, 16:0), arachidic acid (SFA, 20:0), linoleic acid (PUFA, 18:2) and gondoic acid (MUFA, 20:1), as summarized in Figure 5C,D and Table 1. 

Next, we crossed transcriptomics and lipidomics data obtained from TRα- and TRβKO muscles by using Kyoto Encyclopedia of Genes and Genomes (KEGG) mapping (https://www.genome.jp/kegg, accessed on 30 March 2022), in order to identify molecular interactions/relations between the differentially expressed genes and the identified lipidic metabolites. Interestingly, we observed three different protein network interactions, involving GOT2, PGC1-α and UCP3 proteins, which are crucial for mitochondrial function and FAs distribution and transportation (Figure 6A). Based on the fold change in lipidic metabolites’ concentrations, a gene-compound integrated analysis allowed us to determine that all identified compounds in TRs deficient muscle, have altered gene expression linked to FAs and alanine/aspartate/glutamate metabolism. Considering that PGC1-α and UCP3 have already been proved as THs-target genes [36,37], we asked if GOT2 is a novel THs-target gene. To this aim, we performed a Chromatin Immuno-Precipitation (ChIP) assay, which confirmed that the TRs–THs complex physically binds the GOT2 promoter (Figure 6B,C). To further confirm the THs-dependent GOT2 expression, C2C12 cells were treated with THs (30.0 nM T3 and 30.0 nM T4). Interestingly, we observed that THs increase GOT2 expression in a time-dependent manner (Figure 6D). This effect was strongly reduced when C2C12 cells were cultured in THs-deprivation condition (Charcoal Serum) (Figure 6E). Indeed, we observed a significant decrease in GOT2 mRNA expression compared with the cells cultured in Normal Serum, and this reduction was restored after THs treatment (Figure 6E). Together, the data reported above demonstrate that GOT2 is a new THs-target gene in skeletal muscle cells and that the two TRs isoforms exert distinct dysregulation of lipid metabolism in the skeletal muscle, but in both the TRs-deficient conditions, the muscle-specific GOT2 expression drastically drops, causing lipid disorders and oxidative stress.

## 3. Discussion

THs are major metabolic regulators, giving rise to a wide range of effects on growth and development [33,38,39,40,41,42]. The coordinated biological mechanisms by which THs regulate energy metabolism have been recognized for more than 100 years ago, but the key regulatory pathways under THs control still remain to be discovered. 

THs’ most important modus operandi is stimulation or inhibition of gene transcription, achieved through the binding of its active form, T3, to the nuclear receptors. Skeletal muscle is an important target of THs action, which play a crucial role in regulating the metabolism of all the classes of macronutrients [43]. Moreover, it is also well established that THs have a structural regulation, affecting muscle fiber-type characteristics and mitochondrial activity [42,44,45].

THs excess induces a shift toward fast-twitch muscle fiber type [46], whereas hypothyroidism leads to the inverse slow-twitch muscle phenotype [47]. The effects of THs on muscle are mediated mostly by TRα1 [19,48,49,50]. 

In the different THs-target tissues, the TRs proteins display varying expressions both developmentally and spatially, underling a specific tissue-dependent role for each TR isoform (Figure 7). 

Studies in animal models with TRs mutations or treated with TRs agonists have been crucial to clarifying the roles of the two different TR isoforms in the central and peripheral regulation of metabolism by THs. Several studies revealed that, while TRβ is essential to regulate cholesterol metabolism, TRα is necessary for THs-mediated stimulation in energy expenditure and the associated increase in body temperature [27,51,52,53,54]. Suppression of THs signaling by the deficiency of TRα causes a strong down-regulation of several key factors contributing to mitochondrial biogenesis, such as peroxisome proliferator-activated receptor γ coactivator 1α (PGC1α), mitochondrial transcription factor A (TFAM), and estrogen-related receptor α (ERRα) [55]. In our work, we focused on the metabolic changes that occur in skeletal muscle when THs action is impaired by using global knockout TRs mouse models to provide new insight regarding the specific contributions of TRs isoforms on skeletal muscle metabolic phenotype. This study allows several conclusions to be outlined.

In agreement with the general knowledge that the inactivation of TRα isoforms significantly affects normal growth, inducing a lowering of body weights, here, we report that the homozygous inactivation of the *THRα* gene, which abrogates the production of both TRα1 and TRα2 isoforms, exhibits a growth delay and lower muscle fibers size than wild type mice. The absence of TRβ isoforms also leads to lower body weight but has no consequences on muscle fibers morphology and size.

An important observation in the present study is that the TRs deficiency generates mitochondrial dysfunctions. Indeed, we observed: increased mt-ROS generation, reduced expression of genes involved in mitochondrial dynamics, reduced expression of UCP3 and PGC1-α and enhanced intensity of SDH and NADH enzymes. Most of these alterations are indicative of a hypothyroid profile of the TRα- or TRβKO muscles, even though data in the literature suggest that while TRαKO mice are characterized by tissue-specific hypothyroidism, TRβKO mice display a peripheral hyperthyroid state [46,56,57]. Moreover, we observed no changes in mtDNA-CN in TRα-deficient muscles, but a significant increase in mtDNA-CN of TRβ-deficient muscles, which, in the absence of increased mitochondrial function, may represent an adaptive response preceding mitochondrial dysfunction and could therefore be a predictive biomarker of mitochondrial damage [58]. 

Another key finding of the present work is the investigation of the metabolomic profile of skeletal muscle consequent to the single *THRα*- and *THRβ*-LOF. A first observation from the PCA analysis is that both TRα- and TRβKO muscle undergo profound alterations of the lipid composition compared to wild type mice (Figure 5A,B); although, the specific clusterization is suggestive of different alterations in the case of *THRα*- and *THRβ*-LOF. 

The measured intracellular fluctuations in lipids could be attributed to increased β-oxidation occurring in *THRα*-LOF and, on the contrary, to an augmented lipogenesis in *THRβ*-LOF. However, while the described alterations in lipid content of TRα- and TRβKO muscles could be in principle due to muscle-specific metabolic alterations, we cannot exclude that these occur as a consequence of metabolic pathways altered in other tissues, such as the liver or the white adipose tissue.

Considering that the skeletal muscle is not so critically responsible for lipid biosynthesis (compared to, e.g., the liver), and keeping our focus on the skeletal muscle, we matched the metabolomics approach with the transcriptional gene regulation studies in skeletal muscle. 

This match revealed three genes as putative THs-target genes and as potential regulators of lipid metabolism in muscle, namely PGC1-α, UCP3 and GOT2. Strikingly, two of these genes are well-known THs-target genes (PGC1-α and UCP3), while GOT2 expression has not been reported to be regulated by THs. In the literature are described two different functions of GOT2. The much-explored function is as a mitochondrial transaminase, implicated in the maintenance of the malate–aspartate shuttle and redox homeostasis [59]. A second, limited body of evidence suggests a role for GOT2 in metabolite exchange between mitochondria and cytosol, in FAs binding and trafficking, and in facilitating the cellular uptake of long-chain free FAs [60,61]. We investigated the possibility that GOT2 might represent a novel THs-target gene. Indeed, we found that GOT2 mRNA is negatively regulated in TRα- and TRβKO muscles, and ChIP analysis confirmed that the TRs–THs complex physically binds the GOT2 promoter. 

Thus, our study provides the first determination of muscle FAs-transporter GOT2 as a direct THs-target gene and underlines that THs affect FAs oxidation and transport in skeletal muscle. This demonstration could partially explain the lipid disorders and oxidative stress in the two mice models.

Together, the above-reported observations support the concept that the TRs–THs complex in skeletal muscle is a key regulator of mitochondrial bioenergetics and lipid metabolism, and that both TRs are needful for the THs-governed safeguard of the metabolic rate. Moreover, considering that obesity and disorders of lipid metabolism are major health issues, understanding the specific contribution of the TRs isoforms in a tissue-dependent manner could help direct the design and development of THs analogues to treat these disorders.

## 4. Materials and Methods 

### 4.1. Cell Lines, Reagents and Transfection

C2C12 cells were obtained from ATCC and cultured in Dulbecco’s Modified Eagle Medium (DMEM, HiMedia Leading BioSciences Company, Mumbai, Maharashtra, India, cod. AL007) supplemented with 10% Fetal Bovine Serum (FBS, HiMedia Leading BioSciences Company, Mumbai, Maharashtra, India, cod. RM10432), 2.0 mM Glutamine (Gibco, Thermo Fisher Scientific, Waltham, MA, USA, cod. 25030024), 50 i.u. Penicillin/Streptomycin (Gibco, Thermo Fisher Scientific, Waltham, MA, USA, cod. 15070063). C2C12 cells were transfected with CMV-FLAG, TRα-FLAG and TRβ-FLAG plasmids, using Lipofectamine-2000 (Invitrogen™, Carlsbad, CA, USA, cod. 11668019), according to the manufacturer’s instructions. In all the experiments in which THs were applied to cells, we used a combination of T3 (Sigma-Aldrich St. Louis, MI, USA, cod. T6397) and T4 (Sigma-Aldrich St. Louis, MI, USA, cod. T2501) (30.0 nM each, indicated throughout the text and in figures as THs), thus resembling physiological exposure of cells to both the active hormone (T3) and its pro-hormone (T4). In experiments in which THs were removed from the serum, TH-deprivation was achieved by FBS Charcoal absorption.

### 4.2. Mouse Strains

C57BL/6J mice were obtained from Jackson Laboratory (Bar Harbor, ME, USA). The TRα- and TRβKO mice were originally generated by Jacques Samarut (UMR, ENS, Lyon, France) and kindly provided by Graham Williams, Imperial College, London UK, with permission from Dr. Samarut. 12-weeks-old male littermates were used in this study [27,31]. Animals were handled according to national and European Community guidelines, and protocols were approved by the Animal Research Committee of the University of Naples “Federico II” (MIUR, Approval Code: 354/2019-PR).

### 4.3. Animals and Histology

Muscles were dissected and frozen in liquid nitrogen-cooled isopentane, and 7 μm muscle cryosections were used for histology analyses. According to classical methods [62], cryostat sections were stained with Hematoxylin/Eosin (H&E). Briefly, for histology analysis, cross-sections were fixed in 4% formaldehyde at room temperature for 15 min and stained with H&E. Fiber size distribution was quantified by Image-J software (NIH Image, Bethesda, MD, USA). Up to 6 fields of view were captured from the same location within each muscle, and then 600 myofibers/muscle were measured. Images were captured using the Leica Application Suite LAS X Imaging Software with a fluorescent Leica DMi8 microscope.

### 4.4. Histochemical SDH Staining

Histochemical succinate dehydrogenase (SDH) staining was performed on fresh-frozen tissue (FFT). Appropriate cross-sections of muscle were selected and submerged in liquid nitrogen. The tissue was then stored in a −80°C freezer until used. All samples were sectioned on a cryostat at 8–12 μm. Enzymatic activity of SDH was assayed by placing the slides in SDH incubating solution, containing 100.0 mM of sodium succinate salt as a substrate and Nitro-Blue Tetrazolium (NBT) for visualization of reaction, and 1.2 mM of NBT in 0.2 M phosphate buffer for 1 h at 37 °C. Reduced NBT forms a highly colored formazan dye that is finely granular blue. Samples were dehydrated and mounted with Eukitt^®^ mounting medium (Bio-Optica Improving Pathology, Milan, Italy). Images were captured using the Leica Application Suite LAS X Imaging Software with a fluorescent Leica DMi8 microscope and the quantification of SDH activity in muscle was performed using ImageJ software (NIH Image, Bethesda, MD, USA).

### 4.5. Histochemical NADH Staining

The standard Nicotinamide Adenine Dinucleotide (NADH) histochemical staining protocol was followed. Briefly, thawed skeletal muscle cross-sections were incubated in 2.4 mM NADH and Nitro-Blue Tetrazolium (NBT) in 0.5 M Tris buffer for 30 min at 37 °C. Tissues were fixed using 10% phosphate-buffered formalin, washed with a series of acetone solutions, and cover-slipped using Eukitt^®^ mounting medium (Bio-Optica Improving Pathology, Milan, Italy). Images were captured using the Leica Application Suite LAS X Imaging Software with a fluorescent Leica DMi8 microscope and the quantification of NADH activity in muscle was performed using ImageJ software (NIH Image, Bethesda, MD, USA).

### 4.6. Isolation and Loading of Single Skeletal Muscle Fibers with Mito-Sox and Mito-Tracker

12-weeks-old male C57BL/6, TRα- and TRβKO mice were euthanized and the Extensor Digitorum Longus (EDL) muscles were removed and placed into 0.1% Type 1 Collagenase (Sigma-Aldrich St. Louis, MI, USA, cod. C0130) solution in Dulbecco’s Modified Eagle Medium (DMEM, HiMedia Leading BioSciences Company, Mumbai, Maharashtra, India, cod. AL007). Both EDL muscles from each mouse were incubated in collagenase solution for 1 h at 37 °C. Fiber bundles that had not been released during the incubation were separated using a wide-bore glass pipette. The fibers were washed four times in fresh culture medium. Cleaned fibers were plated onto 60 mm dishes in medium for satellite cells (50% DMEM, 50% MCDB, 20% FBS, 1% Ultroser-G (Pall Biosepra, Life Sciences, Cergy-Saint-Christophe, France, cod. 15950-017), 2.0 mM Glutamine, 50 i.u. Penicillin/Streptomycin)). Fibers were incubated for 30 min at 37 °C. The medium was then replaced by PBS containing 5.0 μM Mito-Sox™ Red Mitochondrial Superoxide Indicator (Invitrogen, Thermo Fisher Scientific, Waltham, MA, USA, cod. M36008) or with 200.0 nM Mito-Tracker™ Red CMX-Ros (Invitrogen, Thermo Fisher Scientific, Waltham, MA, USA, cod. M7512) and incubated for 20 min at 37 °C. The fibers were then washed and analyzed by fluorescence microscopy. Images were captured using the Leica Application Suite LAS X Imaging Software with a fluorescent Leica DMi8 microscope and processed using the Photoshop CS5 (Adobe) software package. The Mito-Sox™ and the Mito-Tracker™ integrated density were measured using the ImageJ software (NIH Image, Bethesda, MD, USA). 

### 4.7. Real-Time PCR

Messenger RNAs (mRNAs) were extracted with Trizol reagent (Life Technologies Ltd., Carlsbad, CA, USA, cod. 15596018). Complementary DNAs (cDNAs) were prepared with SuperScript™ VILO™ MasterMix (Life Technologies Ltd., Carlsbad, CA, USA, cod. 11755-050) as indicated by the manufacturer. The cDNAs were amplified by PCR in an iQ5 Multicolor Real-Time Detector System (BioRad, Hercules, CA, USA, cod. 1855201) with the fluorescent double-stranded DNA-binding dye SYBR Green (BioRad, Hercules, CA, USA, cod. 1708882). Specific primers for each gene were designed to work under the same cycling conditions (95 °C for 10 min followed by 40 cycles at 95 °C for 15 s and 60 °C for 1 min), thereby generating products of comparable sizes (about 100–400 bp for each amplification). Primer combinations were positioned whenever possible to span an exon–exon junction and the RNA was digested with DNAse to avoid genomic DNA interference. Primer sequences are reported in Table 2. The relative amounts of gene expression were calculated using Cyclophilin-A (CyA) as the internal standard. All samples were run in triplicate. The results, expressed as N-fold differences in target gene expression, were determined as follows = 2^−(ΔCt target−ΔCt control)^ [63]. For mitochondrial DNA copy number (mtDNA-CN) quantification, 25.0 ng total DNA was used as a template and the RNAaseP gene amplification level were normalized against the nuclear mtND-1 gene as previously described [64].

### 4.8. LC-MS/MS Analysis of Muscle Tissue

Frozen tissue samples were homogenized on ice in Phosphate Buffer, containing 0.25 M sucrose, 1.0 mM EDTA, 0.1 M NaPO4 and 10.0 mM DTT and then sonicated. Proteins were precipitated in Acetonitrile anhydrous, and supernatant evaporated to dryness at 37 °C in a rotavapor and processed for metabolic analysis as previously described [35,65,66]. 

### 4.9. Transcriptomic and Metabolomic Analysis

We analyzed muscle transcriptomics and metabolomics data obtained from skeletal muscle of CTR, TRα- and TRβKO mice. Metabolite set enrichment analysis was performed using the R package MetaboAnalystR [67,68]. Integrated metabolic pathway analysis of transcriptomic and metabolomics data was performed using the same R package.

### 4.10. In Silico Promoter Analysis for Searching Transition Factor Binding Sites

Consensus Thyroid Hormone Receptor Binding Sites (TREs, AC M00239, ID V$T3R_01) with matrix similarity scores of 0.75 or greater (maximum 1.00) in the upstream region of the murine GOT2 gene promoter were identified using TFBIND (http:/tfbind.hgc.jp/, accessed on 30 March 2022). Position analyses of all identified consensus TREs binding sites were reported in Table 3.

### 4.11. Chromatin Immuno-Precipitation (ChIP) Assay

Approximately 2 × 10^6^ C2C12 cells were fixed for 10 min at 37 °C by adding 1% formaldehyde to the growth medium. Fixed cells were harvested, and the pellet was resuspended in 1.0 mL of Lysis Buffer containing protease inhibitors (200.0 mM Phenyl-Methyl-Sulfonyl Fluoride, 1.0 μg/mL Aprotinin). The lysates were sonicated to obtain DNA fragments of 200–1000 bp. Sonicated samples were centrifuged, and the soluble chromatin was diluted 10-fold in Dilution Buffer and used directly for ChIP assays. An aliquot (1/100) of sheared chromatin was further treated with Proteinase K, extracted with phenol/chloroform and precipitated to determine DNA concentration and shearing efficiency (“Input DNA”). Briefly, the sheared chromatin was pre-cleared for 2 h with 1.0 μg of non-immune IgG (Calbiochem, from Sigma-Aldrich, St. Louis, MO, USA) and 30.0 μL of Protein-G Plus/Protein-A Agarose suspension (Calbiochem, from Sigma-Aldrich, St. Louis, MO, USA) saturated with salmon sperm (1.0 mg/mL). Precleared chromatin was divided into aliquots and incubated at 4 °C for 16 h with 1.0 μg of anti-Flag antibody (anti-Flag M2 monoclonal antibody, Sigma-Aldrich, St. Louis, MO, USA, cod. F3165). After five rounds of washing, bound DNA-protein complexes were eluted by incubation with 1% Sodium Dodecyl Sulfate-0.1M NaHCO3 Elution Buffer. Formaldehyde cross-links were reversed by incubation in 200.0 mM NaCl at 65 °C. Samples were extracted twice with phenol-chloroform and precipitated with ethanol. DNA fragments were recovered by centrifugation, resuspended in 50.0 μL H_2_O and used for Real-Time PCRs.

### 4.12. Quantification and Statistical Analysis

As specified in figure legends, the results are shown as the mean ± Standard Deviation (SD). Differences between samples were assessed by the student’s two-tailed t-test for independent samples. GC/MS analysis results were corrected by using the two-way ANOVA test and Bonferroni post-test analysis. Relative mRNA levels (in which the control sample was arbitrarily set as one) are reported as results of Real-Time PCR, in which the expression of Cyclophilin-A (CyA) served as a housekeeping gene. In all experiments, differences were considered significant when the *p*-value was less than 0.05. Asterisks indicate significance at * *p* < 0.05, ** *p* < 0.01, and *** *p* < 0.001 throughout. 

## Figures and Tables

**Figure 1 metabolites-12-00405-f001:**
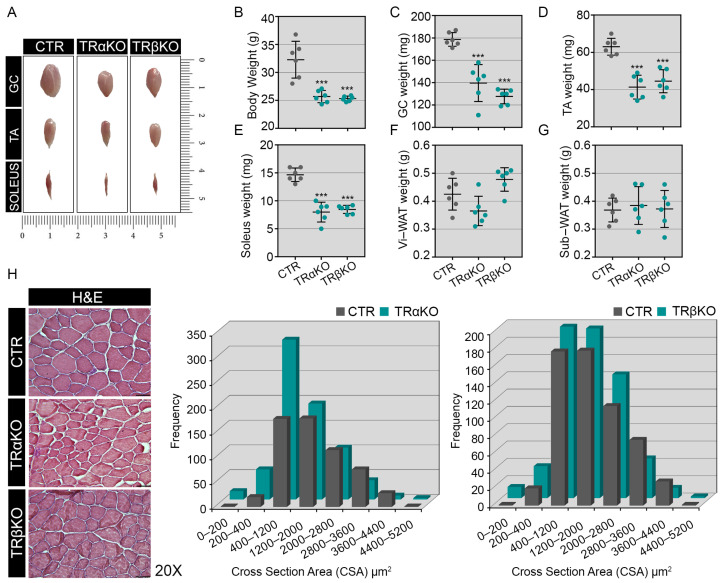
TRα- and TRβKO mice characterization: (**A**) Morphology view of dissected gastrocnemius (GC), tibialis anterior (TA) and soleus muscles of 12-weeks-old TRα-, TRβKO and CTR mice. (**B**–**E**) Analysis of body weight (**B**), GC (**C**), TA (**D**) and soleus (**E**) weight (*n* = 6/group) showed that TRα- and TRβKO mice have a significant reduction in body and lean mass muscle weights compared with CTR littermate. *** *p* < 0.001. (**F**,**G**) Analysis of Visceral White Adipose Tissue (Vi-WAT, B) and Subcutaneous White Adipose Tissue (Sub-WAT, C) showed that TRα- and TRβKO mice have not changed in weights compared with CTR littermate (*n* = 6/group). (**H**) H&E staining and Cross-Sectional Area (CSA) of TA of TRα-, TRβKO and CTR mice. Magnification: 20×. Scale bar: 50 μm.

**Figure 2 metabolites-12-00405-f002:**
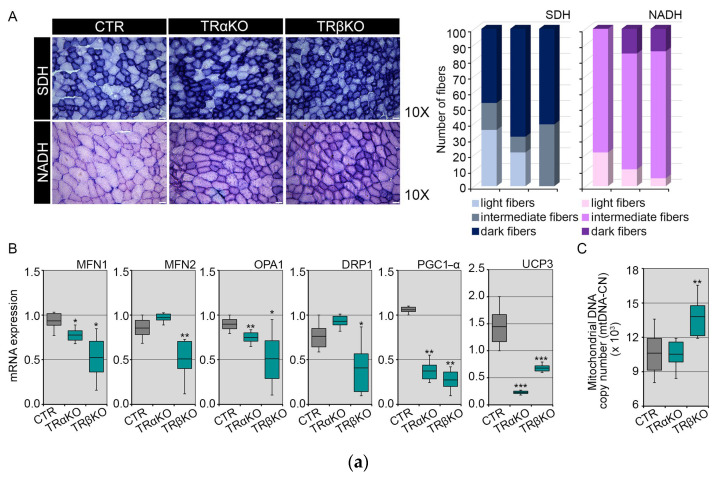
Characterization of mitochondrial dynamic and functionality, in whole-muscle of TRα- and TRβKO mice: (**A**) Succinate dehydrogenase (SDH, top row) and nicotinamide adenine dinucleotide (NADH, bottom row) staining of TA histological sections from TRα-, TRβKO and CTR muscles. CTR (NADH: 21.4% light fibers, 78.6% intermediate fibers and 0% dark fibers; SDH: 35.7% light fibers, 17.1% intermediate fibers and 47.2% dark fibers); TRαKO (NADH: 10.7% light fibers, 89.3% intermediate fibers and 0% dark fibers; SDH: 21.4% light fibers, 10% intermediate fibers and 68.6% dark fibers); TRβKO (NADH: 5% light fibers, 80.7% intermediate fibers and 14.3% dark fibers; SDH: 0% light fibers, 39.2% intermediate fibers and 60.8% dark fibers). Magnification: 10×. Scale bar: 50 μm. Histograms on the right indicate the quantification of SDH and NADH light, intermediate and dark fibers. (**B**) mRNA expression levels of a set of genes involved in mitochondrial biogenesis and turnover (MFN1, MFN2, OPA1, DRP1, PGC1-α and UCP3) were measured by Real-Time PCR in TRα-, TRβKO and CTR mice. Cyclophilin-A was used as an internal control. Normalized copies of the indicated genes in CTR mice were set as 1. Data are shown as mean ± SD (*n* = 6/group). (**C**) Mitochondrial DNA copy number (mtDNA-CN) measured by quantitative Real-Time PCR in TRα-, TRβKO and CTR muscles. Data are shown as mean ± SD (*n* = 6/group). (**D**,**E**) Mitochondrial activity and mitochondrial production of superoxides were measured by Fluorescence-Activated Cell Sorting (FACS) by using Mito-Tracker (**D**) and Mito-Sox (**E**) staining, respectively, on whole-muscle of TRα-, TRβKO and CTR mice. Box Plots on the right show the relative Mean Fluorescence Intensity of TRα-, TRβKO and CTR muscles (*n* = 6/group). The results are shown as means ± SD from at least 3 separate experiments. * *p* < 0.05, ** *p* < 0.01, *** *p* < 0.001.

**Figure 3 metabolites-12-00405-f003:**
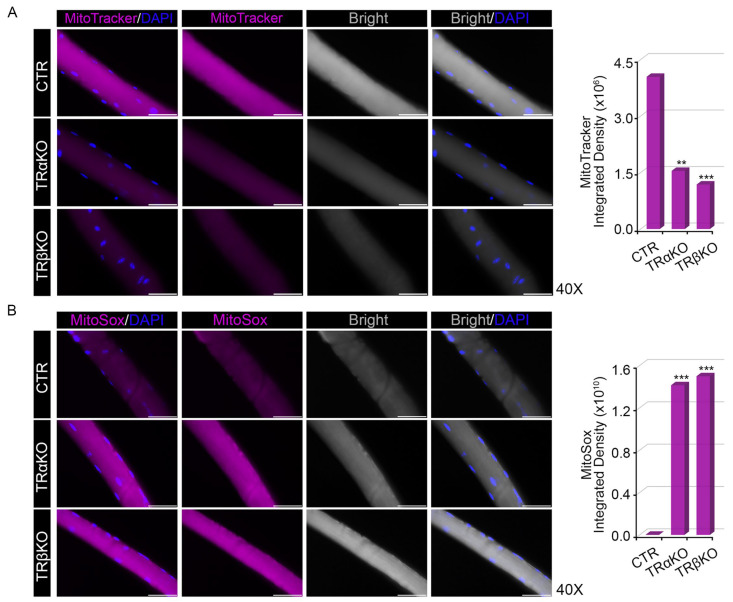
Analysis of mitochondrial features in EDL-derived single muscle fibers isolated from TRα- and TRβKO mice: (**A**,**B**) Mito-Tracker (**A**) and Mito-Sox (**B**) staining on EDL-derived single muscle fibers isolated from TRα-, TRβKO and CTR mice. Magnification: 40×. Scale bar: 50 μm. Histograms on the right indicate the quantification of Mito-Tracker (**A**) and Mito-Sox (**B**) Integrated Density of Fluorescence. ** *p* < 0.01, *** *p* < 0.001.

**Figure 4 metabolites-12-00405-f004:**
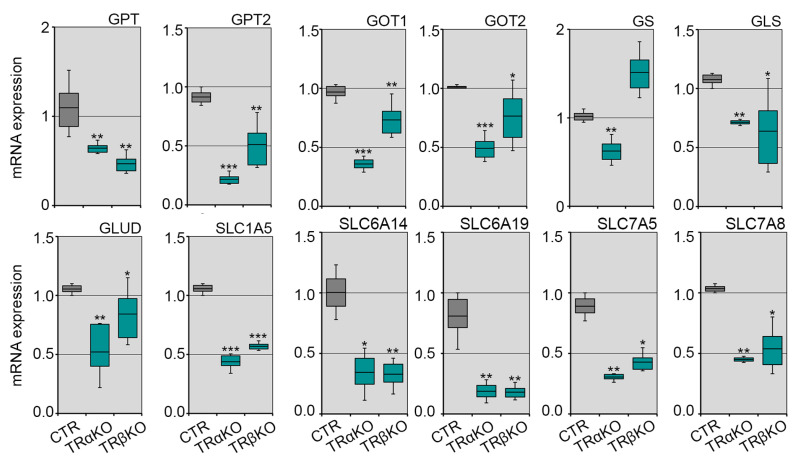
TRs mutations alter muscle physiology by affecting glutamine metabolism: mRNA expression levels of key genes involved in glutamine metabolism regulation (GOT1, GOT2, GS, GLS, GLUD, GPT and GPT2) and glutamine transporters (SLC1A5, SLC6A14, SLC6A19, SLC7A5 and SLC7A8) were measured by Real-Time PCR in TRα-, TRβKO and CTR mice. Cyclophilin-A was used as an internal control. Normalized copies of the indicated genes in CTR mice were set as 1. Data are shown as mean ± SD (*n* = 6/group). * *p* < 0.05, ** *p* < 0.01, *** *p* < 0.001.

**Figure 5 metabolites-12-00405-f005:**
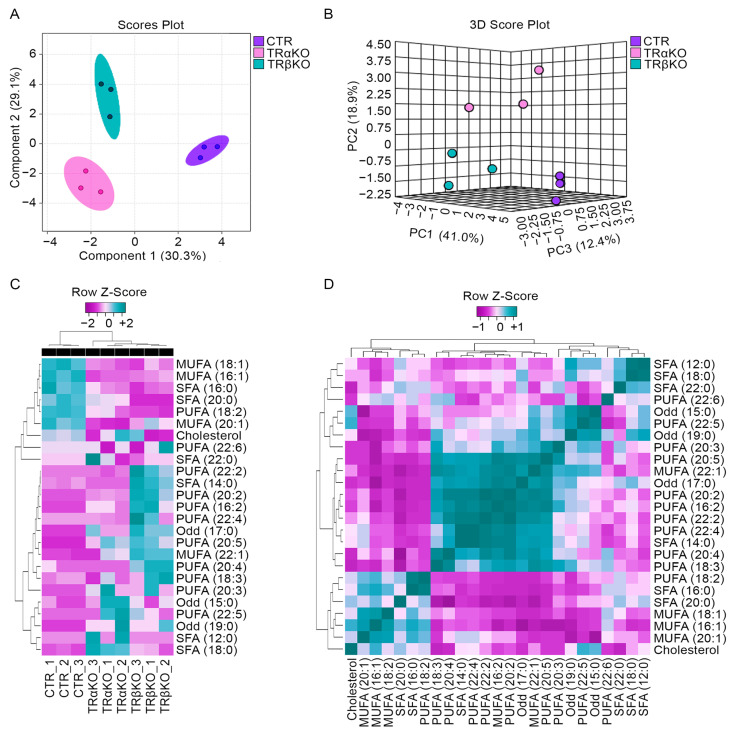
Lipidomics analysis of lipid metabolites in skeletal muscle tissues from TRα-, TRβKO and CTR mice: (**A**) Principal component analysis (PCA) shows the differences in lipid profiles of metabolites extracted from GC muscles of 12-weeks-old TRα-, TRβKO and CTR mice and profiled by GC/MS. Individual data points represent an individual animal. The explained variances are shown in brackets (Component 1 30.3%; Component 1 29.1%). The experiments were run in technical triplicates. (**B**) 3D Score Plot between the selected PCs. The explained variances are shown in brackets (PC1 41.0%; PC2 18.9%; PC3 12.4%). (**C**) Clustering results for metabolites profiled by GC/MS as in (A) are shown as a heat-map. The rows show metabolites, and the columns represent the samples, *p* < 0.001. (**D**) Correlation analysis was used to visualize the overall correlations between the three different groups of mice.

**Figure 6 metabolites-12-00405-f006:**
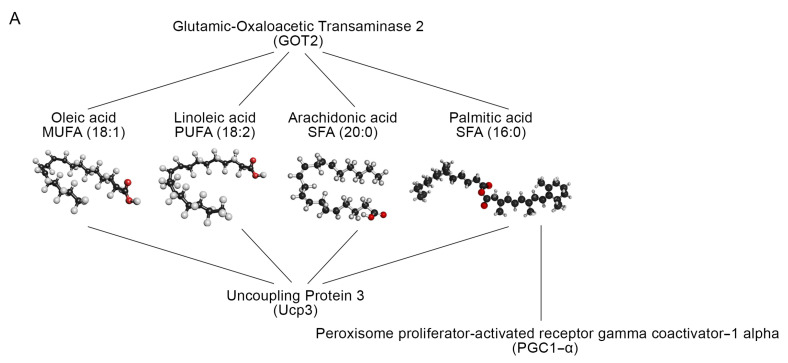
Cross-data obtained by transcriptomics and metabolomics analysis of GC muscle from TRα-, TRβKO and CTR mice: (**A**) Overlapping between FAs/gene-associated networks using KEGG mapping. (**B**) Structure and position from Genome Browser are indicated for the GOT2 promoter. The black box represents the sequence analyzed by ChIP. (**C**) Chromatin Immuno-Precipitation (ChIP) assay was performed in C2C12 cells transiently transfected with a TRα-FLAG and a TRβ-FLAG expressing vector, followed by Real-Time PCR with primers specific for the GOT2 promoter region proximal to TSS. CMV-FLAG plasmid was used as a negative control. The graph shows the Real-Time PCR results with the percentage of chromatin bound. (**D**) GOT2 mRNA expression was evaluated by Real-Time PCR in C2C12 cells treated with THs (30.0 nm T3 and 30.0 nm T4) for 24 and 48 h in Normal Serum. (**E**) GOT2 mRNA expression was evaluated by Real-Time PCR in C2C12 cells cultured in Normal Serum or in TH-depleted medium (Charcoal Serum) ± THs (30.0 nm T3 and 30.0 nm T4) for 24 and 48 h. Data are shown as mean ± SD from at least 3 separate experiments. ** *p* < 0.01, *** *p* < 0.001.

**Figure 7 metabolites-12-00405-f007:**
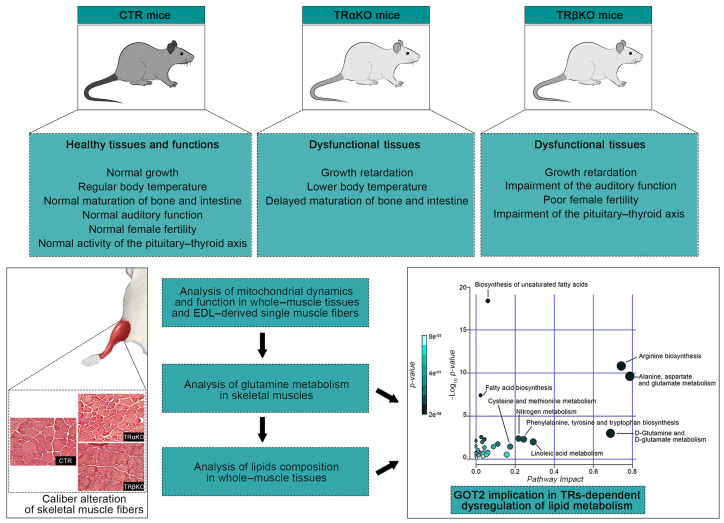
Schematic representation indicating, step by step, the experimental part of the research on the actual state of knowledge about TRs isoforms.

**Table 1 metabolites-12-00405-t001:** List and quantification of metabolites (pmol/μg ± SEM) analyzed by Lipidomics Mass Spectrometry Analysis.

		CTR	TRαKO	TRβKO
Oleic acid	MUFA (18:1)	63.9 ± 1.15	12.6 ± 0.91	14.1 ± 3.10
Palmitoleic acid	MUFA (16:1)	8.1 ± 0.23	1.5 ± 0.31	2.3 ± 0.24
Palmitic acid	SFA (16:0)	71.0 ± 1.35	25.2 ± 1.79	12.9 ± 1.97
Arachidic acid	SFA (20:0)	1.2 ± 0.13	0.8 ± 0.12	0.1 ± 0.00
Linoleic acid	PUFA (18:2)	31.9 ± 0.37	15.4 ± 1.32	6.4 ± 0.56
Cholesterol	Cholesterol	379.1 ± 0.28	366.5 ± 2.64	336.6 ± 2.91
Docosahexaenoic acid	PUFA (22:6)	0.2 ± 0.02	0.2 ± 0.16	0.2 ± 0.25
Behenic acid	SFA (22:0)	0.2 ± 0.02	0.4 ± 0.37	0.2 ± 0.15
Docosadienoic acid	PUFA (22:2)	0.4 ± 0.02	0.4 ± 0.10	1.3 ± 0.32
Myristic acid	SFA (14:0)	2.2 ± 0.18	1.2 ± 0.11	11.4 ± 0.81
Eicosadienoic acid	PUFA (20:2)	0.2 ± 0.02	0.5 ± 0.09	2.1 ± 0.38
Hexadecenoic acid	PUFA (16:2)	0.2 ± 0.02	1.0 ± 0.40	2.7 ± 0.66
Docosatetraenoic acid	PUFA (22:4)	0.3 ± 0.02	0.2 ± 0.25	1.5 ± 1.47
Eicosapentaenoic acid	PUFA (20:5)	0.1 ± 0.01	1.0 ± 0.35	1.3 ± 0.22
Docosenoic acid	MUFA (22:1)	0.3 ± 0.13	0.8 ± 0.71	1.6 ± 0.12
Eicosatetraenoic acid	PUFA (20:4)	0.3 ± 0.12	0.2 ± 0.00	0.5 ± 0.08
Octadecatrienoic acid	PUFA (18:3)	1.1 ± 0.08	1.2 ± 0.79	2.4 ± 0.71
Eicosatrienoic acid	PUFA (20:3)	0.2 ± 0.02	0.5 ± 0.34	0.4 ± 0.30
Docosapentaenoic acid	PUFA (22:5)	0.2 ± 0.02	0.5 ± 0.37	0.3 ± 0.21
Lauric acid	SFA (12:0)	0.2 ± 0.02	0.3 ± 0.12	0.2 ± 0.01
Stearic acid	SFA (18:0)	13.7 ± 0.39	45.0 ± 0.82	18.6 ± 1.08

**Table 2 metabolites-12-00405-t002:** Primers used for Real-Time PCR.

Oligos	Sequences	Amplicon Length
CyA	Forward CGCCACTGTCGCTTTTCG Reverse AACTTTGTCTGCAAACAGCTC	120 bp
DRP1	Forward CTAGAAGAGCCCAGCCTACG Reverse AGCAAAGTCGGGGTGTTTTG	237 bp
GOT1	Forward CTCCTCCGGTTCTGGTCTTT Reverse CCCCAAGAACTAGGCGAGAA	234 bp
GOT2	Forward CAGCCGAGATGTCTTTCTGC Reverse GGACACTCTGCTCTGGGATT	167 bp
GOT2-ChIP	Forward CCCAGAGAGTCTATACAGTTC Reverse GGGGGCCACAGCATGTGCTGC	390 bp
GLS	Forward GCTGTGCTCTATTGAAGTGA Reverse GCAAACTGCCCTGAGAAGTC	175 bp
GLUD	Forward GAGATGTCCTGGATCGCTGA Reverse GGCCCACATTACCAAATCCC	274 bp
GPT	Forward TTCAAGAAGGTGCTCACGGA Reverse CATCTGTTTCTGCACCTCGG	278 bp
GPT2	Forward GCTTTGAATGTGGACGAGCT Reverse TCGTGCCCCATCTGGTAAAG	254 bp
GS	Forward ACAGCGACATGTACCTCCAT Reverse CTGCTCCATTCCAAACCAGG	185 bp
MFN1	Forward GATGACCTGGTTTTAGTAGAC Reverse TGAAGATGTTGGGCTTGGAGA	187 bp
MFN2	Forward TCGAGAGGCAGTTTGAGGAG Reverse GCAGCTTGTAGTCTTGAGCC	187 bp
mtND-1	Forward GTGAGTGATAGGGTAGGTGCA Reverse AACACTCCTCGTCCCCATTC	219 bp
OPA1	Forward GAAGGACGACAAAGGCATCC Reverse AGTCACCTTCACTGGAGAGC	244 bp
PGC1-α	Forward TGATGTGAATGACTTGGATACAGACA Reverse GCTCATTGTTGTACTGGTTGGATATG	95 bp
RNAaseP	Forward CTGACCACACGAGCTGGTAGAA Reverse GCCTACACTGGAGTCGTGCTACT	61 bp
SLC1A5	Forward CGGGACCTCTTCTAGCTCTG Reverse GGACACCCCGTTTAGTTGTG	181 bp
SLC6A14	Forward ATTCCCTTTCTGTGGCTTGG Reverse AGAGCTTCAGGATAGGCAATGA	235 bp
SLC6A19	Forward GGCCTCATCTCCTTCTCCAG Reverse AGTCATCAAAGCGCTCAGTG	154 bp
SLC7A5	Forward CCCTGGCCCTCATCATTTTG Reverse GAAGAGGCCGCTGTACAAAG	149 bp
SLC7A8	Forward CAGCGCCTGTGGTATCATTG Reverse AGACTTAGGGATGGTGACGC	181 bp
UCP3	Forward GCCTACAGAACCATCGCCAG Reverse GCCACCATCTTCAGCATACA	302 bp

**Table 3 metabolites-12-00405-t003:** Consensus sequence analysis of the 24 different putative motifs for Thyroid Hormone Receptor (AC M00239, ID V$T3R_01) within the GOT2 promoter region.

AC	Score	Loc.	Strand	Consensus	Sequence
M00239	0.77	10	(−)	SNNTRAGGTCACGSNN	ACTGCTGACCTAACTC
M00239	0.73	33	(+)	SNNTRAGGTCACGSNN	CCTTATGGTCAATTAT
M00239	0.82	352	(+)	SNNTRAGGTCACGSNN	GGATAAGGTCATATTA
M00239	0.73	364	(−)	SNNTRAGGTCACGSNN	ATTAGTGACTCCATCT
M00239	0.75	409	(+)	SNNTRAGGTCACGSNN	ACTTAAGATCTTGGCC
M00239	0.76	543	(−)	SNNTRAGGTCACGSNN	CTCAGTCGCCTCAGTC
M00239	0.74	552	(−)	SNNTRAGGTCACGSNN	CTCAGTCGCCTCAGTT
M00239	0.74	618	(+)	SNNTRAGGTCACGSNN	CAACAAGCTCAAAGCA
M00239	0.81	642	(+)	SNNTRAGGTCACGSNN	CTGTAAGCTCATTATA
M00239	0.75	653	(+)	SNNTRAGGTCACGSNN	TTATAAGTTCAAAGCA
M00239	0.73	705	(−)	SNNTRAGGTCACGSNN	ACCTGTGGCTTCATCC
M00239	0.75	783	(+)	SNNTRAGGTCACGSNN	ATGAAAGCTCACTGTA
M00239	0.73	935	(+)	SNNTRAGGTCACGSNN	CTGTGAAATCACTAAT
M00239	0.77	1467	(+)	SNNTRAGGTCACGSNN	GTTTGAGGCCAGTCTG
M00239	0.75	1490	(+)	SNNTRAGGTCACGSNN	AAGTGAGTTCCAGGAC
M00239	0.75	1611	(+)	SNNTRAGGTCACGSNN	GGTTAAGAGCACTGTC
M00239	0.74	1633	(+)	SNNTRAGGTCACGSNN	TCCAGAGGTCATGAGT
M00239	0.75	1639	(−)	SNNTRAGGTCACGSNN	GGTCATGAGTTCAATT
M00239	0.77	1773	(+)	SNNTRAGGTCACGSNN	GCCCAAGGTCAGGGAG
M00239	0.73	1845	(−)	SNNTRAGGTCACGSNN	GATCTGGACTTCATGC
M00239	0.83	2045	(−)	SNNTRAGGTCACGSNN	ATCCATCACCTTATAG
M00239	0.78	2166	(−)	SNNTRAGGTCACGSNN	TAGTGTGGCCACATTG
M00239	0.77	2283	(+)	SNNTRAGGTCACGSNN	GATTAAAGGCATGCAC
M00239	0.78	2357	(−)	SNNTRAGGTCACGSNN	ACCAGTGACCTGGGCC

## Data Availability

We thank the CEINGE–Biotecnologie Avanzate Scarl, Naples for help with the Flow cytometry experiments.

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
