# Peer review of "Thyroid Hormone Receptor Isoforms Alpha and Beta Play Convergent Roles in Muscle Physiology and Metabolic Regulation"

_metabolites, 2022, doi:10.3390/metabo12050405_

Round 1
Reviewer 1 Report
The aimof the current study is to investigate the muscle metabolic response to TRs abrogation. The authors performed many functional analysis very well elucidated in the method section. The results are original and innovative. However, some points deserve more attention to improve the presentation of the paper.
1- In the introduction the authors dwelt on the mechanism of action of thyroid hormone, which is very well known and described before, without specifying the problem of the study relating to muscle metabolism, to physiopathological notions...
2- The results were not properly discussed. The discussion was shortened and limited to brief reporting of results without deep projection into physiological mechanisms or pathophysiological contexts.
Author Response
The aim of the current study is to investigate the muscle metabolic response to TRs abrogation. The authors performed many functional analysis very well elucidated in the method section. The results are original and innovative. However, some points deserve more attention to improve the presentation of the paper.
1) In the introduction the authors dwelt on the mechanism of action of thyroid hormone, which is very well known and described before, without specifying the problem of the study relating to muscle metabolism, to physiopathological notions...
A. We thank the reviewer for the suggestion, we have added in the Introduction an additional explanation of the aim of our study and its implications in the pathophysiology of TH action in skeletal muscle (pages 2 and 3).
2) The results were not properly discussed. The discussion was shortened and limited to brief reporting of results without deep projection into physiological mechanisms or pathophysiological contexts.
A. We extended the description of the Results as requested.
Reviewer 2 Report
The enclosed manuscript is a well-organized and presented study which focused on the characteristics of THRa and THRb in skeletal muscle functions, particularly mitochondrial metabolism. The authors used transgenic mice with respective TH receptors knocked out and studied the gene and physiological changes. Later, mitochondrial functions were highlighted, followed by systemic de novo findings of target genes and putative mechanisms. The flow of research works was straightforward and reasonable, despite a few issues being noticed.
1) In fig 2D and 2E, both were unclearly distributed on the flowcytometric results. With reference to the significant changes on the box plot (right), it's a bit confusing how the authors converted the flow cytometric histogram to the box plot, or maybe the authors did another quantitative measurement. Moreover, the figure legend may have some misleading info, e.g., (D, E) ... mito-tracker (C) and mito-sox (D)...
2) I'm sort of conserved on the use of "gene mutation" in the 3.2 subtitle. Apparently, the authors were using a knockout model instead of a specific mutational pattern on THRa and THRb genes.
3) It was a great effort from the authors to elucidate the route of mechanism in the mitochondria-dependent muscle control; however, there was no further proof-of-concept studies to accomplish that GOT2 can dominate the mitochondrial metabolic system activation as well as the skeletal muscle differentiation. It is understandable that the authors may not want to over-expand the contents; however, it's a bit of pity for missing that part.
4) It is noted that TRa is less dominant compared to TRb. The authors may want to further discuss if TRa involves other energy consumption and metabolic systems.
Author Response
The enclosed manuscript is a well-organized and presented study which focused on the characteristics of THRa and THRb in skeletal muscle functions, particularly mitochondrial metabolism. The authors used transgenic mice with respective TH receptors knocked out and studied the gene and physiological changes. Later, mitochondrial functions were highlighted, followed by systemic de novo findings of target genes and putative mechanisms. The flow of research works was straightforward and reasonable, despite a few issues being noticed.
1) In fig 2D and 2E, both were unclearly distributed on the flowcytometric results. With reference to the significant changes on the box plot (right), it's a bit confusing how the authors converted the flow cytometric histogram to the box plot, or maybe the authors did another quantitative measurement. Moreover, the figure legend may have some misleading info, e.g., (D, E) ... mito-tracker (C) and mito-sox (D)...
- We apologize for not having clearly described Figures 2D and E. We used only the Mito-Sox and Mito-Tracker as quantitative analysis. In Cytometric histograms, cell count is reported, and we evaluated the probe positive cells population; next, we have calculated the mean fluorescence intensity of six different samples for each genotype, using unstained control as a negative reference.
2) I'm sort of conserved on the use of "gene mutation" in the 3.2 subtitle. Apparently, the authors were using a knockout model instead of a specific mutational pattern on THRa and THRb genes.
- We agree with the reviewer and apologize for the mistake. The Title has been corrected.
3) It was a great effort from the authors to elucidate the route of mechanism in the mitochondria-dependent muscle control; however, there was no further proof-of-concept studies to accomplish that GOT2 can dominate the mitochondrial metabolic system activation as well as the skeletal muscle differentiation. It is understandable that the authors may not want to over-expand the contents; however, it's a bit of pity for missing that part.
- We thank the reviewer for raising this point, further additional experiments will be addressed to deepen the role of GOT2 in skeletal muscle. In the literature is reported that GOT2, a member of the malate-aspartate shuttle (MAS), plays an essential role in the intracellular NAD(H) redox balance and it is also involved in metabolite exchange between mitochondria and cytosol and in fatty acid-binding and trafficking (Yang, H., Embo J, 2015; Yang, S., Cell Death Dis, 2018). The two different functions of GOT2 depend on its subcellular compartmentalization. Functionally, in muscle, it appears that GOT2 is a fatty acid transport protein on the plasma membrane, and GOT2 is a functional enzyme at the level of the mitochondria, facilitating the transport of reducing equivalents into the mitochondrial matrix (Graham P Holloway, J Physiol. 2007 Jul) and regulating the rate of the reaction glutamate + oxaloacetate ⇌ aspartate + 2-oxoglutarate in mitochondria (Lehninger et al. 1993). Our findings show that the muscle FAs-transporter GOT2 is a direct THs-target gene and underlines that THs affect FAs oxidation and transport in skeletal muscle. Further studies will be designed by our group to elucidate the role of GOT2 at the mitochondrial level and during the myogenic process.
4) It is noted that TRa is less dominant compared to TRb. The authors may want to further discuss if TRa involves other energy consumption and metabolic systems.
- We agree with the reviewer, and we have included an additional explanation of the metabolic function of the TRa isoforms in the Discussion section (pag. 11)
Reviewer 3 Report
Dear Authors,
I consider the submitted article for my review to be good and consistent.
The title and the abstract reflect the content of the research. The introduction is insightful and shows the actual state of knowledge about thyroid hormone receptor isoforms alpha and beta. The experimental part of the research is the strong point of this article. The methods are adequate, well-chosen, and well explained. To make it easier for the reader to get through this part of the article - I suggest adding a scheme here which would facilitate (step by step) following the experimental part of the research.
The results were presented with due diligence and insight, and I have not noticed any abuses in this regard. The figures are of good quality and well described. I consider both their quantity and quality to be appropriate. I am thinking about Tables 1 and 2 - whether they should be included in the article or the appendix. I leave it for possible consideration of the authors. The discussion in the manuscript is good. I think it could be helpful to elaborate on more conclusions. In addition, I propose to add a limitation of the studies and, finally, I recommend the indication of further research possibilities in this area and an indication of how the obtained results can be used.
A summary diagram would also be helpful here. The authors would present the metabolic pathways and indicate what they managed to detect here, what their novelty is, and what, for example, is not yet known here.
Author Response
Dear Authors,
I consider the submitted article for my review to be good and consistent. The title and the abstract reflect the content of the research. The introduction is insightful and shows the actual state of knowledge about thyroid hormone receptor isoforms alpha and beta. The experimental part of the research is the strong point of this article. The methods are adequate, well-chosen, and well explained. To make it easier for the reader to get through this part of the article - I suggest adding a scheme here which would facilitate (step by step) following the experimental part of the research.
The results were presented with due diligence and insight, and I have not noticed any abuses in this regard. The figures are of good quality and well described. I consider both their quantity and quality to be appropriate. I am thinking about Tables 1 and 2 - whether they should be included in the article or the appendix. I leave it for possible consideration of the authors. The discussion in the manuscript is good. I think it could be helpful to elaborate on more conclusions. In addition, I propose to add a limitation of the studies and, finally, I recommend the indication of further research possibilities in this area and an indication of how the obtained results can be used. A summary diagram would also be helpful here. The authors would present the metabolic pathways and indicate what they managed to detect here, what their novelty is, and what, for example, is not yet known here.
A. We thank the reviewer for the suggestion. We have moved the table 1 and 2 after the discussion section and we have added another figure (Figure 7) that summarizes the main concepts of our work.